# Assessment of the Physical, Mechanical, and Tribological Properties of PDMS Thin Films Based on Different Curing Conditions

**DOI:** 10.3390/ma14164489

**Published:** 2021-08-10

**Authors:** Gang-Min Kim, Sung-Jun Lee, Chang-Lae Kim

**Affiliations:** Department of Mechanical Engineering, Chosun University, Gwangju 61452, Korea; khb417@naver.com (G.-M.K.); k3668591@chosun.kr (S.-J.L.)

**Keywords:** curing, friction, heat transfer, PDMS, ultrasonic vibration, wear tracks

## Abstract

Polydimethylsiloxane (PDMS), a silicone-based elastomeric polymer, is generally cured by applying heat to a mixture of a PDMS base and crosslinking agent, and its material properties differ according to the mixing ratio and heating conditions. In this study, we analyzed the effects of different curing processes on the various properties of PDMS thin films prepared by mixing a PDMS solution comprising a PDMS base and a crosslinking agent in a ratio of 10:1. The PDMS thin films were cured using three heat transfer methods: convection heat transfer using an oven, conduction heat transfer using a hotplate, and conduction heat transfer using an ultrasonic device that generates heat internally from ultrasonic vibrations. The physical, chemical, mechanical, and tribological properties of the PDMS thin films were assessed after curing. The polymer chains in the PDMS thin films varied according to the heat transfer method, which resulted in changes in the mechanical and tribological properties. The ultrasonicated PDMS thin film exhibited the highest crystallinity, and hence, the best mechanical, friction, and wear properties.

## 1. Introduction

Polydimethylsiloxane (PDMS) is a silicon-based elastomeric polymer used as the base material in various advanced devices for research, owing to its advantageous optical, chemical and mechanical properties, easy manufacturing, and excellent formability [1]. In the past decades, numerous studies have used PDMS as the substrate in various applications, such as piezo electrodes, flexible electrodes, stress-strain sensors, pressure sensors, triboelectric nanogenerators due to its intrinsic elasticity and excellent mechanical properties, in biomedical devices considering it is harmless to the human body and does not induce chemical reactions during solidifying, and in circuit boards, micro-fluid systems, and MEMS due to its high formability and transparency [2,3,4,5,6,7,8,9,10,11].

Additionally, besides enhancing the inherent characteristics of PDMS for use as substrates, researches have also been conducted to reinforce or alter the intrinsic properties of PDMS to facilitate its use as a circuit board. Furthermore, research has been conducted to analyze the variations in surface properties using plasma surface treatment (or surface patterning), such as PDMS sponge using excellent oil hygroscopicity, and PDMS elastomer fabrication through gamma-ray radiation [12,13,14].

Commercial PDMS elastomers are prepared by mixing a PDMS base and a curing agent in a ratio of 10:1, followed by heat curing, which results in the formation of a silyl hybrid group due to contact between a vinyl or allyl group and a silicone-based material [15,16,17]. Both the PDMS base and curing agent harden while mixing to form a network structure of chemical bonds between the PDMS base and curing agent by crosslinking. However, considering the crosslinked polymer is similar in shape to a crystalline structure, a 100% bonding between the crosslinking agent and polymer base is not possible [17]. The chemical reaction speed of the thermosetting curing agent differs according to the changes in temperature, given that chemical reactions arise due to heat curing. In addition, given that the inherent properties of the PDMS change based on its molar molecular weight, the mechanical properties of PDMS vary depending on the mixing ratio of the curing agent and base material, as well as the heating temperature, and curing time [18,19,20,21]. Moreover, although many studies have been conducted to control the mechanical properties of solidified PDMS by adding micro/nano-sized materials [22,23,24], identifying the optimal condition to grow PDMS elastomers that exhibit specific characteristics is challenging.

The mechanical properties of a polymer depend on the degree of its crystallization. In a previous study, it has been reported that the degree of crystallization of polymers is closely related to wear resistance, and the elastic modulus and yield stress of polymers with high crystallinity are usually high [25]. As such, it means that the mechanical properties of polymers can be changed by controlling the degree of crystallization in polymeric bonding. The curing method of PDMS involves adding a crosslinking agent to the PDMS base and then applying heat. The variation in the curing method affects the bonding between the molecules inside the polymer, resulting in a change in crystallinity. In this study, changes in the physical, mechanical, and tribological properties of PDMS specimens prepared by different curing methods, despite the fact that the PDMS base and curing agent were mixed at the same constant ratio, were analyzed.

## 2. Materials and Methods

### 2.1. Materials

The commercial Sylgard184 product (DC-184, Dow Corning, Seoul, Korea), used in various research fields, was used to prepare for PDMS base samples and the crosslinking agent. The PDMS thin film specimen was prepared using the fabrication method shown in Figure 1. First, the PDMS base and crosslinking agent were mixed in a ratio of 10:1, stirred, and left at room temperature (20–25 °C) for 30 min to eliminate air bubbles. Then, using the dip-coating method, a thin film was formed from the prepared solution using a stainless-steel plate with a smooth surface (Ra: 0.3 μm). The coating on the stainless-steel plate was solidified in a conventional way using an oven and a hot plate and also cured using an ultrasonication device (Frequency: 40,000 Hz). The resultant product was PDMS thin films. The difference in the curing methods is determined by the convection and conduction heat transfer methods, which use the oven and hotplate, respectively. The ultrasonication curing method hardens the PDMS using ultrasonic waves with conduction heat transfer by generating vibration in the PDMS through physical stimulation. The oven, hotplate, and ultrasonication device were preheated to 80 °C, and the liquid PDMS thin films were cured for 1 h. During the cooling process of the cured PDMS thin film specimens, the temperature of each curing device was allowed to cool gradually to around 20–25 °C to prevent the specimens from deforming due to temperature change. As a result, three different PDMS thin film specimens were prepared, which showed no visual differences considering all the other parameters, except the curing method, were fixed. The PDMS thin film specimens were approximately 500 µm thick and were cut into pieces of the same size (35 mm × 35 mm).

### 2.2. Experiments

To analyze the surface properties of the fabricated PDMS thin film specimens, the water contact angle and surface roughness were measured. After a 10 µL water droplet was dropped on the surface of the PDMS thin film using a micropipette, the contact angle of the water droplet was observed using a microscope camera (U1000X, Wtong Industry Group, Seoul, Korea). The surface roughness of the PDMS thin film was measured using a 3D laser scanning confocal microscope (3D-LSCM, VK-X200, KEYENCE Co. Osaka, Japan). The measurements were repeated over three times to compare and obtain an average value before further evaluation.

The components of the PDMS base before and after curing were analyzed according to the curing method using attenuated total reflectance (ATR-Nicolet 6700, Thermo Scientific, Waltham, MA, USA) and Fourier transform infrared spectroscopy (FT-IR Spectrum 400, Perkin Elmer, Waltham, MA, USA), respectively. ATR and FT-IR are quantitative and qualitative analysis methods that use the intrinsic spectra of substances by measuring their absorbance based on each component of the organic/inorganic molecular compounds in the infrared region. Precisely, these methods were used in this study to analyze the changes in the internal components of PDMS that are contingent on the crosslinking agent, and compare the changes according to the curing method.

The crystallization degree of the PDMS thin films according to the curing method was analyzed using the X-ray diffraction analysis method (XRD-X’Pert Pro MRD, Malvern PANalytical, Malvern, Worcestershire, UK). Based on the measurement results, the crystallite sizes of each thin film specimen were compared.

Furthermore, the mechanical properties of the PDMS thin films, such as stiffness, elastic modulus, shear modulus, and bulk modulus, were analyzed using a custom-built micro-indentation tester. While applying a load vertically to the PDMS thin film fixed on the stage, the changes in the force according to the amount of deformation exhibited by the specimens were measured through the load cell in real-time. A 25.4 mm steel ball was used as the indenter tip and pressed in the surface by 20% of the PDMS thin film thickness to minimize the influence of the substrate. In order to accurately measure the change of reaction force according to the indentation depth, the loading/unloading operation was performed at a constant speed of 0.05 mm/s as slowly as possible. Then, the mechanical properties of each specimen were evaluated by analyzing the measured indentation force-depth data. In order to verify the reliability of the experimental results, all experiments were repeated three or more times under the same conditions.

The friction properties of the PDMS thin films were evaluated using a micro-tribotester (RFW160, Neoplus.co, Daejeon, Korea). A certain load was applied from the contact between the steel ball tip and the PDMS thin film surface fixed to the reciprocating stage, which induced a relative sliding motion between the two contact surfaces, and the magnitude of the friction force was measured in real-time [26]. The friction and wear properties of the PDMS thin film specimens were analyzed by varying the total sliding repetitions under different contact pressures using different tip diameters and vertical loads. Under mild friction conditions, the steel ball tip was 2 mm wide, the normal load was 50 mN, and the reciprocating sliding motion was performed for 120 cycles. Conversely, under severe friction conditions, the diameter of the steel ball was reduced to 1 mm, the normal load was doubled to 100 mN, and the reciprocating sliding motion was performed for 10,000 cycles. The contact pressure was calculated using the Hertzian contact theory under the two friction conditions (0.5 MPa for mild and 1.0 MPa for severe friction conditions), which confirmed the difference of two times between both friction conditions. The friction test conditions are listed in Table 1. The wear tracks formed on the PDMS thin films after the experiment was conducted were analyzed using a 3D laser scanning confocal microscopy (3D-LSCM, VK-X200, KEYENCE Co. Osaka, Japan) to calculate the wear rates of the PDMS thin films according to the curing conditions.

## 3. Results and Discussion

The main hypothesis of this study is that the physical, mechanical, and tribological properties of PDMS differ owing to the difference in the heat transfer mechanisms. Generally, both heat transfer methods, the convection method using an oven and the conduction method using a hot plate, transfer heat from the outside to the PDMS solution. However, they have a different mechanism for heat transfer in the PDMS solution, wherein convection is used to heat the PDMS solution by transferring heat from the surface to the inside, whereas conduction is used to heat from the bottom of the PDMS solution due to which heat is transferred to the inside. Moreover, when the ultrasonication device is used, heat is transferred from the outside to the PDMS using the conduction mechanism, wherein heat and vibration are simultaneously generated between the polymeric molecules inside the PDMS due to ultrasonic waves.

To analyze the surface properties of the PDMS thin films, the surface roughness and contact angle of the water droplets were compared, as seen in Figure 2. The surface roughness (R_a_) of the three specimens showed slight differences ranging from 0.12–0.15 µm. In the oven curing method, which uses the convection heat transfer mechanism, the external surface of the thin film hardened, and the surface roughness value was measured as approximately 0.12 µm. In the hotplate curing method, which uses conduction heat transfer mechanism, the PDMS thin films were cured starting from the bottom. After the specimen was cured and the heat was transferred, surface roughness of approximately 0.15 µm was observed. In the ultrasonication curing method, which uses the conduction heat transfer mechanism, the bonding of molecules inside the PDMS films was altered due to the ultrasonic vibration, and the surface roughness value was measured as approximately 0.12 µm. As mentioned above, it is believed that the voids and non-uniformities within the PDMS films change depending on the curing process, which in turn affects the surface roughness. The contact angles of water drops measured on the PDMS thin films were found to be approximately 108°. Considering the surface roughness of the specimens was different but the contact angles were almost the same, it was confirmed that the hydrophobicity in PDMS is a chemical property of the polymer component. It is also believed that the different surface roughness did not significantly affect the contact angle, that is, the hydrophobicity of the surface. 

Molecular spectroscopy analysis was used to measure changes in the internal chemical structure and crystallinity of the PDMS thin films. Figure 3 shows the ATR/FT-IR spectra and XRD pattern results. Figure 3a shows a graph of the ATR analysis result of the liquid PDMS base in the initial state without the curing agent, which was compared to the untreated and ultrasonicated liquid PDMS base. The main characteristic bands of the ATR spectra for the untreated and ultrasonicated PDMS base showed a similar trend as that of the previous study [27]. 

Regardless of the ultrasonic treatment, the intrinsic wavelength of each PDMS base was as follows: asymmetric stretching vibration was observed due to the C-H stretching of CH_3_ in the 2950–2960 cm^−1^ infrared band range, CH_3_ symmetrical bending in the 1260 cm^−1^ infrared band, Si-O-Si asymmetric and symmetric stretching in the 1020–1074 cm^−1^ infrared band range, absorption peaks of -CH_3_ rocking, and Si-C stretching in Si-CH_3_ in the 789–796 cm^−1^ infrared band range [27]. The intensity of the peak was slightly higher in the major intrinsic band of the ultrasonicated PDMS compared to that of the untreated PDMS base, which was revealed from the Si-O-Si bonding of the 1020–1074 cm^−1^ infrared band of the ultrasonicated PDMS base. It was the result that the entangled PDMS chains were detangled due to the dispersion of the ultrasonic waves, and it affected the crystallization of PDMS [27,28].

As shown in Figure 3b, differences were also observed in the results of the FT-IR spectra analysis of the PDMS thin films. Based on the IR spectrum analysis of the hardened thin film, the difference appeared in the 1020–1074 cm^−1^ and 789–796 cm^−1^ infrared bands, where each peak shows an Si-O-Si and Si-CH_3_ bonding, the main chemical bonds of the hardened PDMS, indicating that the degree of internal bonding between the PDMS base and curing agent has increased [29,30]. The peak of the PDMS film cured using ultrasonic treatment was the strongest, while it weakened in order in the oven and hotplate methods, respectively. This behavior was due to the decrease in the distance between the chains considering the tight bonding between the PDMS base and curing agent, which reduces the crystallite size of the cured PDMS and increases the crystallization degree [31].

Figure 3c shows the results of the XRD analysis of the PDMS thin films. To measure the crystallization degree and the crystallite size, the FWHM of the strong peak was calculated using the Gaussian function. The crystallization degree was compared with the relative quantity of crystal between the specimens based on the FWHM value, and the crystallite size was measured using Debye–Scherrer’s equation based on the FWHM value and peak position. The crystallite size was measured as
(1)$$\tau =\frac{K\lambda }{\beta \cos\theta }$$
where *τ* is the crystallite size, *β* is the FWHM value, cos*θ* is the Bragg angle, and *K* is the dimensionless shape coefficient (a constant value that varies depending on the crystallite size, but generally values ranging from 0.89–0.9 are used), and *λ* is the wavelength of the X-ray light source (using a wavelength of 1.542 Å). The crystallization degree and the crystallite size determined from the Gaussian function and Scherrer’s equation, respectively, are compared in Table 2.

From the XRD measurement, the XRD peaks’ 2*θ* positions of three different PDMS thin films were observed to be are adjacent to 11.8°, which indicates the minicrystal in the PDMS specimens [32]. Considering the crystallization degree of PDMS at 2*θ* = 11.8°, the FWHM value of the PDMS specimen cured using ultrasonication, oven, and hot plate methods were 2.981, 2.672, and 2.572°, respectively, that is, highest and lowest for ultrasonication and hot plate methods. It means that the crystallite size increases in the order of the PDMS specimens cured by ultrasonication, oven and hot plate methods. The PDMS thin film cured by ultrasonication has the smallest crystallite size of approximately 2.682 nm because the reaction between the PDMS base and the curing agent is activated to form a dense crosslinking as the chains of randomly entangled polymeric molecules are detangled due to the ultrasonic wave stimulation. Additionally, the oven-cured specimen, which is cured simultaneously over a large area by heat injection from all directions on the surface and sides, has higher crystallinity than the hot plate-cured specimen in which the heat is conducted only in one direction. Furthermore, molecular spectroscopy analysis confirmed that the degree of internal chemical bonding and crystallinity of the PDMS specimen cured by ultrasonication was the highest. Figure 4 illustrates the hypothesis of this study that ultrasonic stimulation detangles long molecular chains with the large molecular weight of the polymer and cuts the copolymer block.

The mechanical properties of the PDMS thin films were evaluated using Oliver and Pharr’s theory and further analyzed using the indentation experiment data [33]. Among the various formulas depending on the shape of the indenter tip, we utilized the elastomer specimen formula [33,34]. Table 3 shows the stiffness, elastic modulus, shear modulus, and bulk modulus values of each specimen calculated using the Oliver & Pharr theory from the results of the indentation experiments conducted on the specimens [34,35,36]. It was confirmed that the mechanical properties of the PDMS thin films used in this study are almost similar to the mechanical properties of PDMS suggested in previous studies, which show different values according to the curing method [18,19,20,21]. The mechanical properties of the ultrasonicated PDMS thin films showed the best results. Additionally, the ATR, FT-IR, and XRD analyses verified the changes in the polymer chain and highest crystallinity in the PDMS film after ultrasonic treatment. As such, it can be explained that the mechanical properties of the ultrasonicated PDMS thin film, whose crystallite size was measured as the smallest, were improved.

In the reciprocating-type sliding friction experiments, the friction coefficients of the PDMS thin films were measured under mild and severe friction conditions, which are summarized in Table 1. As observed, there was approximately a factor of two between the contact pressures of the PDMS thin films under the two friction conditions. The average friction coefficients of the PDMS thin films according to the curing conditions in each experimental condition were compared, as seen in Figure 5. Under the mild friction condition, the average friction coefficients of the PDMS thin film cured in the oven and the hot plate showed similar values of 1.54 and 1.50, respectively, while the PDMS thin film cured using ultrasonication showed the lowest friction coefficient of approximately 1.03. Furthermore, the friction coefficient under severe friction conditions increased slightly to approximately 1.57 in the oven-cured specimen, approximately 1.67 in the hot plate-cured specimen, and approximately 1.06 in the ultrasonicated specimen, as compared to the mild condition. Particularly, for the hot plate-cured specimen, the increase rate of the friction coefficient was the highest at 11% and lower for ultrasonic waves by over 30% as compared to other heat transfer principles, because the surface stiffness is higher owing to the high crystallinity of the ultrasonicated PDMS thin film, which was confirmed from the ATR/FT-IR analysis results. In addition, because the amount of indentation deformation is relatively small under the same load, it is assumed that the friction force is lowered owing to the decrease in the real contact area. This can be verified by comparing the mechanical properties of the PDMS thin films, as listed in Table 3.

As shown in Figure 6a, we compared the wear rates of the PDMS thin films according to the curing conditions. The damaged part is defined as the part of the wear track that rises above the surface of the PDMS due to repeated sliding motion [37,38]. The wear volume of the damaged part was measured by 3D-LSCM analysis. The wear rate was calculated using the same equation used in previous studies [39], given as
(2)$$Wear\;rate\;\left(WR\right)=\;\frac{Wear\;volume\;\left(V\right)}{Normal\;load\;\left(L\right)\times Sliding\;distance\;\left(D\right)\;}\;$$

The normal load (L) was constant at 100 mN, whereas the total sliding distance (D) was equal to 40,000 mm during the entire slide cycle of 10,000 cycles, considering it is a wear track length with 2 mm stroke and moves 4 mm through the reciprocal slide. According to Equation (2), the wear rates of PDMS thin films cured using the oven, hot plate, and ultrasonic methods were calculated as 2.92 × 10^−7^ mm^3^/N·mm, 3.92 × 10^−7^ mm^3^/N·mm, and 1.88 × 10^−7^ mm^3^/N·mm, respectively. Furthermore, it was confirmed that the degree of wear rate was inversely proportional to the size of the mechanical properties (stiffness, elastic modulus, shear modulus, and bulk modulus values) of the PDMS thin films analyzed through the indentation experiments. The wear rate was lowest in the ultrasonication (vibration + conduction heat transfer) method, followed by the oven (convection heat transfer) and hotplate (conduction heat transfer) methods, indicating excellent wear resistance. In other words, by comparing the mechanical properties of each specimen, such as stiffness, elastic modulus, shear modulus, and bulk modulus, we determined that wear occurs the least in the PDMS specimen cured by ultrasonication resulting in the lowest wear rate, owing to its excellent mechanical properties. After the friction experiment, the morphology of the wear track formed on the surface of each specimen was observed. As shown in Figure 6b, the surface of the wear track was raised above the original surface as a whole and partially scratched in all PDMS specimens. The degree of damage was highest on the surface of the hot plate-cured specimen, followed by the oven-cured and ultrasonicated specimens.

As shown in Figure 7, when the steel ball tip comes in contact with the PDMS surface, which is a silicon-based elastomer, and a normal load is applied, the contact area on the PDMS surface is pressed down and rises from around the spherical tip. Due to the initial surface deformation, the contact area increases, leading to a high overall friction coefficient of the PDMS thin film based on sticking phenomenon that obstructs the steel ball tip from sliding, which is normal, owing to the intrinsically high adhesive property of the PDMS material. As seen in Figure 7ac, there were differences in the initial contact states of the cured PDMS thin films. Based on the difference in the degree to which the steel ball tip was pressed into the PDMS thin film surface, it is understood that the mechanical properties of each specimen differ. As a result, the PDMS thin film cured by ultrasonication, exhibiting the highest stiffness and elastic modulus, was pressed the least, whereas the PDMS thin films cured by the oven or hotplate methods, which exhibited relatively low mechanical properties, was pressed more. The pressed depth of the hotplate-cured PDMS thin film, having the lowest mechanical properties, was expected to be the highest.

Under the friction conditions, the PDMS thin films showed a plowing wear phenomenon-based initial wear pattern, a wear mechanism mainly found in elastomers. When the PDMS thin film surface was pressed by the ball tip and the contact surface was deformed, the surface rose in the front of the tip sliding direction, resulting in a large friction force due to the sticking phenomenon. After the tip passes the surface, the risen surface returns to its original flat state owing to the elastic properties of the PDMS. However, contact and frictional stress accumulate on the contact surface due to repeated sliding movements, resulting in the surface cracking and bursting due to fatigue, indicating that the strongly bonded polymer chains were broken and rose above the surface. Therefore, it was assumed that the wear surfaces of all the cured PDMS thin films had risen; however, the degree of protrusion or scratching on the damaged surface differed according to the curing method. While the ultrasonicated PDMS thin film showed the least traces of damage, the hot plate-cured thin film showed the most severe traces of damage. The degree of damage was observed to be consistent with the tendency of the mechanical properties of PDMS thin films based on the curing methods. In other words, the mechanical properties of the ultrasonicated PDMS thin film with the least damage were the best. According to the ATR analysis result, there was a distinct difference in the Si-O-Si bonding of the ultrasonicated PDMS in a specific band, indicating changes in the internal polymer chains. Additionally, the FT-IR analysis results showed that the main characteristic peaks of the Si-O-Si and Si-CH_3_ bonds of the ultrasonicated PDMS thin film were the strongest. It was observed the internal bond between the PDMS base and curing agent increased. However, owing to the tight bonding between the PDMS base and curing agent, the distance between the chains decreased resulting in a decreased the crystallite size and increased the crystallization degree in the cured PDMS thin films, considering the crosslinked PDMS comprises dense bonds formed due to the bond between loose chains rather than bonded chains, owing to the molecular structure. Conversely, when curing with only heat transfer (oven or hot plate methods), the entangled polymer chain and crosslinking agent form bonds, resulting in a relatively large crystallite size and a minimal crystallization degree as compared to the ultrasonication method. Moreover, the difference in the crystallization degrees of the oven-cured and hot plate-cured thin films is due to the different heat transfer mechanisms (convection and conduction). As a result, the main characteristic band peak of ATR decreased in the order of ultrasonication, oven, and hotplate methods, the internal bonding weakened, the crystallite size increased, and the crystallization degree decreased, resulting in weakened mechanical properties. Accordingly, it is believed that the degree of damage increases. The curing agent used in this study is a thermosetting curing agent, which actively forms chemical bonds when cured at high temperatures above 80 °C. While curing in the oven-cured specimen undergoes heat transfer in all directions, the hot plate-cured specimen undergoes one-way heat transfer, resulting in a higher crystallization degree of the oven-cured over the hot-plate-cured specimen, thereby indicating that the higher the crystallinity, the better are the mechanical and tribological properties of the PDMS thin film, which was verified through XRD analysis. The crystallinity of the ultrasonicated PDMS specimen was the highest and exhibited the strongest internal chemical bonding, whereas its crystallite size was the smallest, assuming the combined PDMS base and curing agent mixture spread equally in all directions due to the effect of ultrasonic waves, thereby loosening the polymer chains that cause mechanical vibration between the two components. The crystallinity increased and crystal size decreased in order in the oven and hotplate methods. Overall, the results achieved for the ultrasonicated PDMS thin film may have contributed to reinforcing the mechanical properties of the PDMS thin film, resulting in decreased damage to the specimen. Furthermore, if and when damage occurs by the propagation of contact pressure and frictional stress in the PDMS specimen due to repeated contact sliding motion, the durability of the ultrasonicated PDMS thin film is considered excellent owing to the strong internal polymer bonding. In other words, the ultrasonicated PDMS thin film with high crystallinity exhibited a strong surface strength, and the wear deformation caused by friction was the least. Additionally, the degree of wear increased in the oven-cured and hot plate-cured specimens in order. In other words, the ultrasonicated PDMS thin film with high crystallinity exhibited a strong surface strength, and the wear deformation caused by friction was the least [25]. Furthermore, based on the results of experiments conducted to determine the mechanical strength and surface tribological properties of PDMS films according to the crystallinity, it was observed that the higher the crystallinity was, the better the tribological properties and mechanical strength were in case of polymers such as PDMS.

## 4. Conclusions

In this study, we prepared different kinds of PDMS thin films by the convective heat transfer method using an oven, the conductive heat transfer method using a hotplate, and an ultrasonic wave/conductive heat transfer method using an ultrasonication device. The variations in the physical, mechanical, and tribological properties of the PDMS thin films according to the curing method were analyzed. The ultrasonication method showed the most ideal characteristics, followed by the oven and hotplate methods, respectively. From the ATR, FT-IR, and XRD analyses, the crystallinity of the specimen cured using ultrasonic waves was observed to be the highest with the smallest crystallite size. As a result, the PDMS specimen cured by simultaneously applying ultrasonic waves and conductive heat transfer were more densely crystallized than those cured using the conventional oven and hotplate curing methods, thereby resulting in excellent mechanical properties and friction/wear characteristics. Accordingly, the curing conditions affecting the chemical bonding between the PDMS base and the crosslinking agent play an important role in the physical, mechanical, and tribological properties of the PDMS thin film.

## Figures and Tables

**Figure 1 materials-14-04489-f001:**
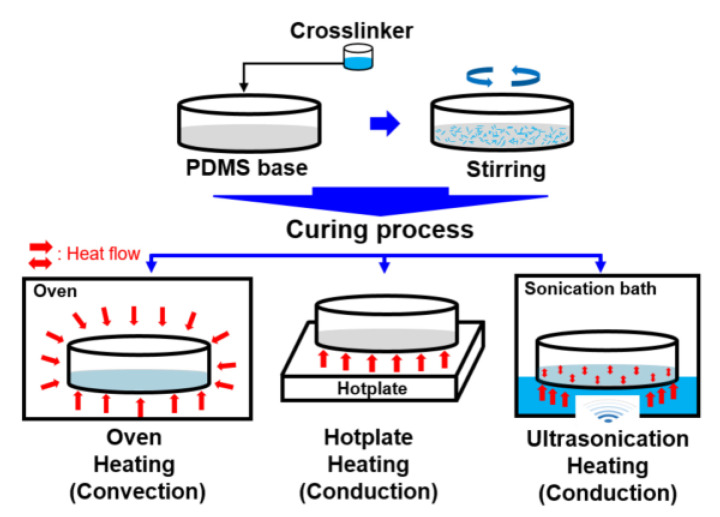
Different fabrication methods of the PDMS films: oven heating, hotplate heating, and ultrasonication heating.

**Figure 2 materials-14-04489-f002:**
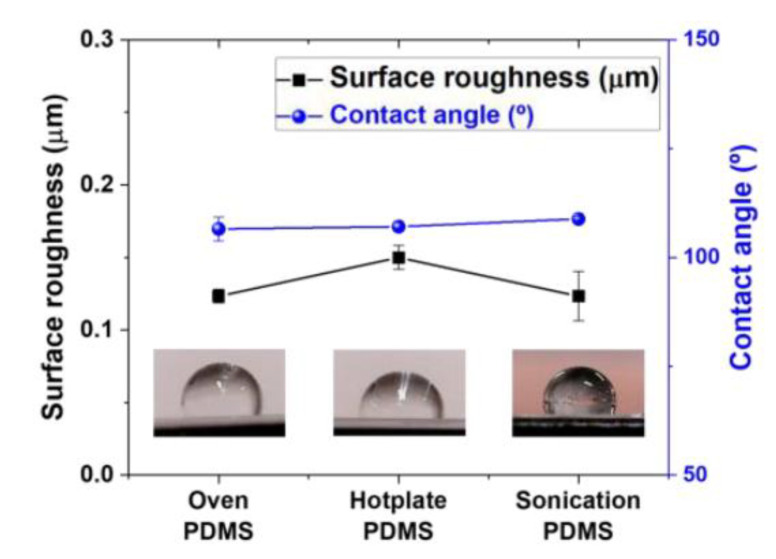
Surface roughness and contact angles of water droplets on PDMS thin films. Insert images: Microscopic camera images of water droplets on the surface of the PDMS thin films under different curing methods.

**Figure 3 materials-14-04489-f003:**
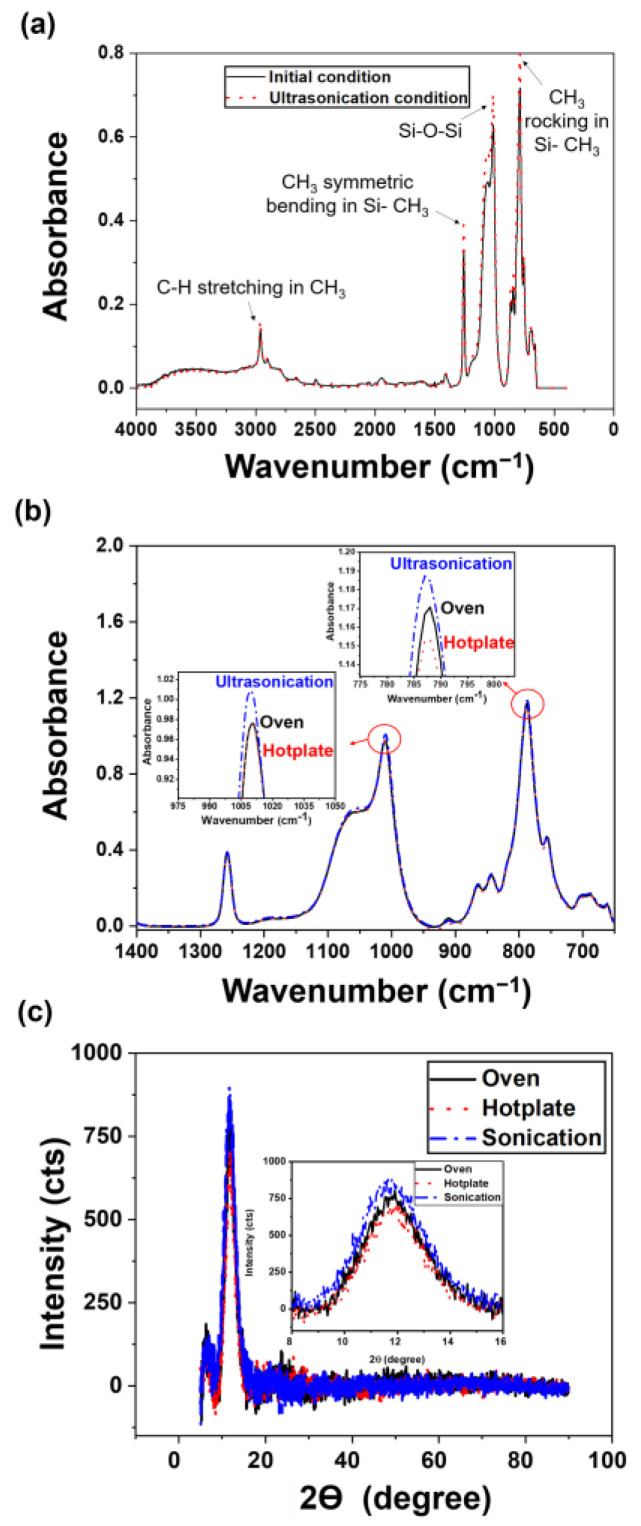
Molecular spectroscopy analysis. (**a**) ATR spectra of the PDMS base solution with and without ultrasonic treatment, (**b**) FT-IR spectra, and (**c**) XRD pattern of the PDMS thin films under different curing methods.

**Figure 4 materials-14-04489-f004:**
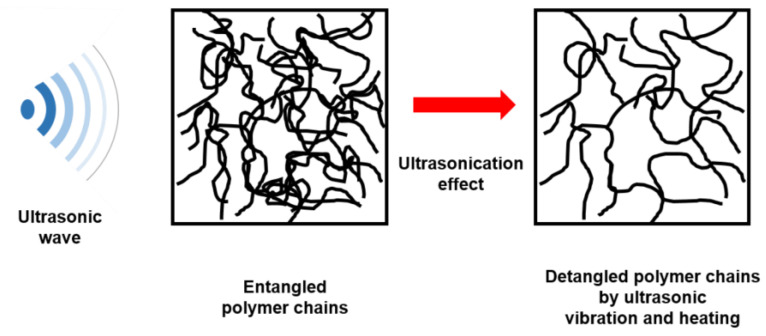
Variation in polymer chains due to ultrasonication.

**Figure 5 materials-14-04489-f005:**
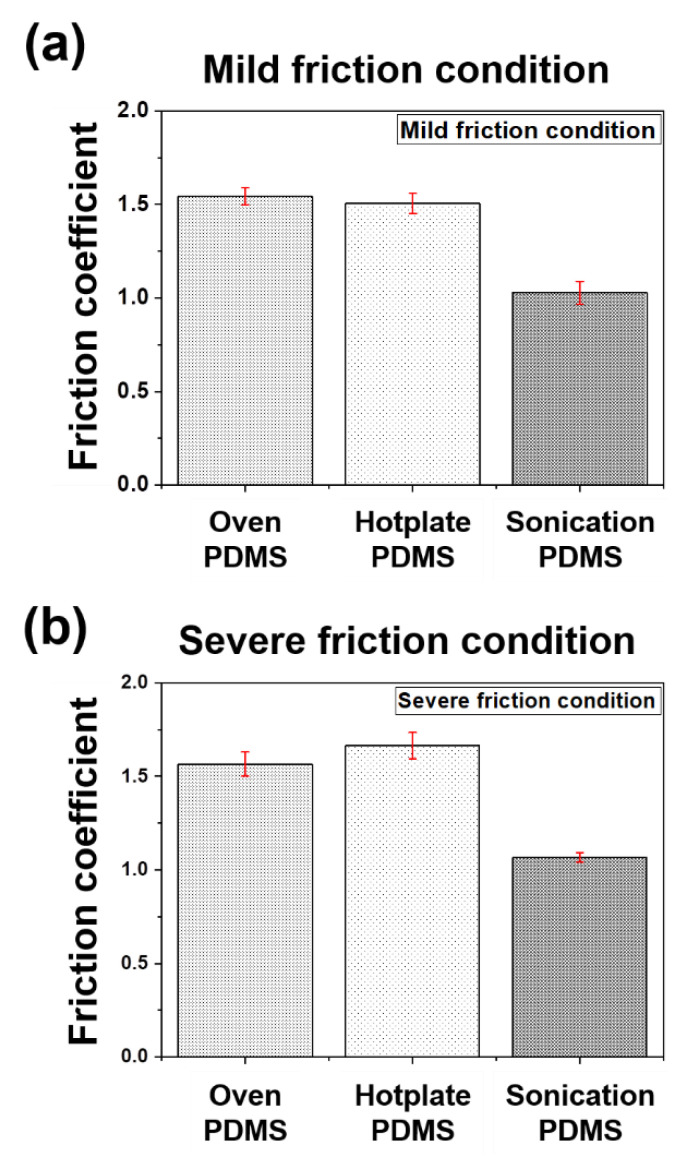
Average friction coefficients of the PDMS thin films under different curing methods. (**a**) mild friction condition (contact pressure: 0.5 MPa, sliding cycles: 120 cycles); (**b**) severe friction condition (contact pressure: 1 MPa, sliding cycles: 10,000 cycles).

**Figure 6 materials-14-04489-f006:**
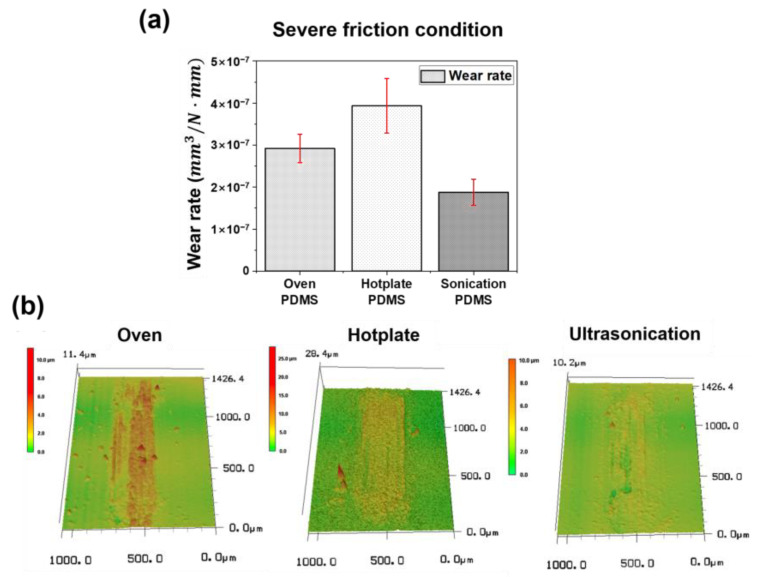
(**a**) Average wear rates, and (**b**) 3D laser scanning confocal microscope images of the wear tracks of the PDMS thin films during different curing methods after sliding test under 1 MPa contact pressure during 10,000 sliding cycles.

**Figure 7 materials-14-04489-f007:**
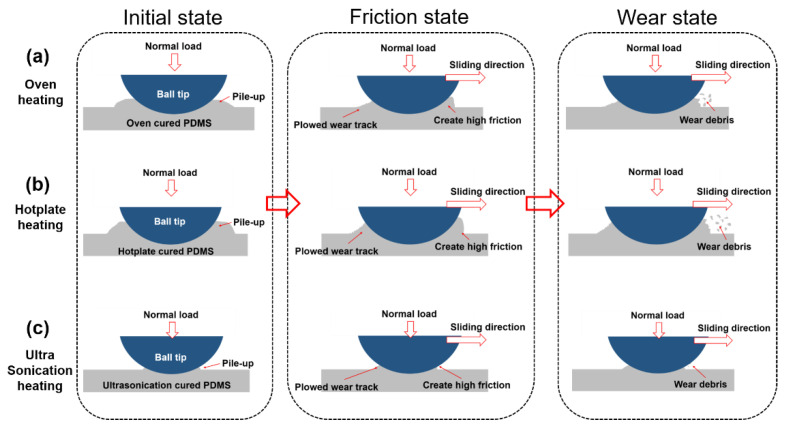
Initial contact state and friction/wear mechanism of the PDMS thin films cured by the (**a**) oven, (**b**) hotplate, and (**c**) ultrasonication curing methods.

**Table 1 materials-14-04489-t001:** Experimental conditions of the friction test.

Variables	Friction Conditions	
	Mild	Severe
Tip type	Steel ball	Steel ball
Tip diameter (mm)	2	1
Normal load (mN)	50	100
Contact pressure (MPa)	0.5	1
Sliding cycles	120	10,000

**Table 2 materials-14-04489-t002:** FWHM and crystallite size of the PDMS thin films under different curing methods.

XRD Data Analysis	Peak Position (2θ)	FWHM (°)	Crystallite Size, τ (nm)
Oven	11.869	2.672	2.991
Hotplate	11.879	2.572	3.109
Ultrasonication device	11.806	2.981	2.682

**Table 3 materials-14-04489-t003:** Mechanical properties of the PDMS thin films under different curing methods.

Curing Method	Stiffness (N/mm)	Elastic Modulus (MPa)	Shear Modulus (MPa)	Bulk Modulus (GPa)
Oven	7.43 ± 0.06	2.34 ± 0.04	0.78 ± 0.01	0.39 ± 0.006
Hotplate	7.17 ± 0.19	1.93 ± 0.16	0.64 ± 0.05	0.32 ± 0.026
Ultrasonication device	7.94 ± 0.03	2.61 ± 0.003	0.87 ± 0.001	0.44 ± 0.001

## Data Availability

Data is available on request from the corresponding author.

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
