# Peer review of "Assessment of the Physical, Mechanical, and Tribological Properties of PDMS Thin Films Based on Different Curing Conditions"

_materials, 2021, doi:10.3390/ma14164489_

Round 1

Reviewer 1 Report

This manuscript could be accepted after making some supplement as following:
1.    What is the background and purpose of this study, what is the potential application?
2.    What is the crosslinking agent? What kind of reaction happened between PDMS and the crosslinking agent?
3.    Is the crosslinking agent reacted all in each state?

Reviewer 2 Report

The authors have discussed the effect of various curing methods of on the mechanical, physical, chemical, and tribological properties of PDMS. In the introduction, the authors have given the detailed literature study on the various methods to enhance intrinsic, surface properties of PDMS for different applications. They have explained the basic hypothetical mechanism of heat transfer in different curing methods and its effect of physical, mechanical and tribological properties of PDMS. According to this research work, the overall physical, mechanical and tribological properties of the PDMS films showed better properties when cured using ultrasonic waves followed by the oven and hot plate curing methods. I find this study very interesting and conducted in thorough manner and can be accepted after improving the quality of Figures. I still have one following question and a suggestion:

  1. Why did the authors choose PDMS and crosslinking agent mixing ratio 10:1? Though commercially the mixing ratio is used 10:1 between PDMS and curing agent, the authors could have used different mixing ratios to effectively understand the effect of these curing methods on PDMS properties.
  2. I would suggest redrawing most figures as there is no uniformity in figures texts fonts size. All the figures quality and presentation must be improved.

Reviewer 3 Report

The article is certainly interesting: it focuses on the properties variation of Polydimethylsiloxane (PDMS) films while changing the heat treatment modality. Roughness, wettability, mechanical and tribological properties of PDMS coatings on smooth steel are evaluated. However, in my opinion it needs an overall revision, as it is a bit confusing and unclear. Some illustrative comments follow.

ABSTRACT

Line 16

The word "on" should be replaced with "after".

INTRODUCTION

Line 18

I would delete the word "However".

Line 38

What is used as curing agent?

Lines 53-54

I ask the authors the courtesy to rewrite this sentence as it is not understandable to me.

Line 55-64

This part is also a bit confusing. It is not clear what has been done, what are the doubts to be clarified and what the authors have come up with to try to explain them.

MATERIALS

Line 79

Simulation?

Lines 81-83

Below room temperature?

Line 86

How did the authors evaluate the thickness?

EXPERIMENTS

Line 94

The word "was" should be removed.

Lines 110-111

I believe "collison" is not the proper word to use.

Lines 117-120

How did the authors choose these test conditions? Which method did you use? It should be described here.

RESULTS AND DISCUSSION

Line 223-232

I do not understand what ref [27] has to do with the 11.8° XRD peak the authors found.

“However, the opposite tendency was observed for the crystallite size.” This behavior is intrinsic in the Debye – Scherrer formula.

“The PDMS thin films cured by ultrasonication showed the smallest crystallite size of approximately 2.725 nm, considering the chains between polymeric molecules were detangled due to the ultrasonic waves, causing mechanical vibration, and the 231 bonding between the PDMS base and the curing agent was equal in all directions.” What do the authors mean?

Line 234

“it is cured in a relatively wide range”. What do the authors mean?

Line 240-244

“As shown in Fig. 4, the ultrasonic stimulation in this study not only detangled long molecular chains due to the large molecular weight of the polymer but also cut the copolymer block, indicating that the mechanical and tribological properties of the PDMS thin film were enhanced with an increased crystallization degree of the PDMS specimen cured by ultrasonication.”

Neither these considerations, nor Figure 4 are in the right place. They serve to explain what hypothetically happens. Furthermore, mechanical and tribological properties have not yet been discussed.

Line 247-275

Please do not enter Oliver and Pharr's method: just add a couple of references.

Line 336-337

“As shown in Fig. 7(b), the surface of the wear track was raised above the original surface as a whole and partially scratched in all PDMS specimens.” and subsequent lines. It is not at all clear to me what happens and how the wear rate is evaluated. What would be the eroded volume?

Reviewer 4 Report

The authors have discussed the effect of three different curing methods and their influence on the mechanical, physical, chemical, and tribological properties of the formed thin films. The basic mechanisms of the different heat transfer methods are well described as well as their effect on the various properties of the thin films.
The authors conclude that the crystallization degree increases and the crystallite size decreases by using the ultrasonication method. This leads to an overall improvement in physical, mechanical and tribological properties of the PDMS thin films.

In spite of using only simple improvements of already established methods which are leading to expected results, the study could be interesting for the community for further improvement and use of such PDMS thin films.

The study is carried out thoroughly and can be accepted after minor adjustments to spelling and grammar.

Line 25 change: [1] -> non superscript
Line 66 change: materials -> Materials
Line 71 change: 25 °C -> 25 °C
Line 190 change: CH3 -> CH3
Line 190 change: cm-1 -> cm-1
Line 191 change: cm-1 -> cm-1
Table 2 change: D -> τ
Line 228 change 2.572 -> 2.572 nm
Line 237 change: most ideal -> highest
Line 261 change: Eq. -> eq.
Line 264 change: Eq. -> eq.
Line 293 change: difference of two times -> factor of two
Line 323 change: Eq. -> eq.

Round 2

Reviewer 3 Report

The authors have greatly improved the quality of the manuscript.

Author Response

Thank you for your valuable review of our paper.